# Immunogenicity and Safety of the BNT162b2 COVID-19 Vaccine in Patients with Cystic Fibrosis with or without Lung Transplantation

**DOI:** 10.3390/ijms24020908

**Published:** 2023-01-04

**Authors:** Francesca Lucca, Valentino Bezzerri, Elisa Danese, Debora Olioso, Denise Peserico, Christian Boni, Giulia Cucchetto, Martina Montagnana, Gloria Tridello, Ilaria Meneghelli, Mirco Ros, Giuseppe Lippi, Marco Cipolli

**Affiliations:** 1Cystic Fibrosis Center, Azienda Ospedaliera Universitaria Integrata di Verona, 37126 Verona, Italy; 2Section of Clinical Biochemistry, Department of Neurological, Biomedical and Movement Sciences, University of Verona, 37134 Verona, Italy; 3Unit of Pediatrics, S. Maria di Ca’ Foncello Hospital, 31100 Treviso, Italy

**Keywords:** cystic fibrosis, BNT162b2, COVID-19, SARS-CoV-2

## Abstract

Cystic fibrosis (CF) is characterized by a progressive decline in lung function, which may be further impaired by viral infections. CF is therefore considered a comorbidity of coronavirus disease 2019 (COVID-19), and SARS-CoV-2 vaccine prioritization has been proposed for patients with (pw)CF. Poor outcomes have been reported in lung transplant recipients (LTR) after SARS-CoV-2 infections. LTR have also displayed poor immunization against SARS-CoV-2 after mRNA-based BNT162b2 vaccination, especially in those undergoing immunosuppressive treatment, mostly those receiving mycophenolate mofetil (MMF) therapy. We aimed to determine here the immunogenicity and safety of the BNT162b2 vaccine in our cohort of 260 pwCF, including 18 LTR. Serum levels of neutralizing anti-SARS-CoV-2 IgG and IgA antibodies were quantified after the administration of two doses. PwCF displayed a vaccine-induced IgG and IgA antiviral response comparable with that seen in the general population. We also observed that the immunogenicity of the BNT162b2 vaccine was significantly impaired in the LTR subcohort, especially in patients undergoing MMF therapy. The BNT162b2 vaccine also caused minor adverse events as in the general population, mostly after administration of the second dose. Overall, our results justify the use of the BNT162b2 vaccine in pwCF and highlight the importance of a longitudinal assessment of the anti-SARS-CoV-2 IgG and IgA neutralizing antibody response to COVID-19 vaccination.

## 1. Introduction

A new zoonotic disorder (Coronavirus disease 2019; COVID-19) caused by the severe acute respiratory syndrome coronavirus 2 (SARS-CoV-2) emerged in 2019, stirring up worldwide attention and concerns due to millions of confirmed deaths worldwide. The pandemic proportions led to a huge global response to this outbreak, resulting in the accelerated development of vaccines against SARS-CoV-2. The spike (S) glycoprotein of this coronavirus plays a crucial role in viral entry into the host cells. This protein is composed of two main subunits, the S1 region, which includes the receptor binding domain (RBD) for recognition of angiotensin-converting enzyme 2 (ACE2) on the host cell surface, and the S2 subunit, which is instead essential for fusion of the viral envelope with the cellular membrane. Antibodies against the S protein, in particular targeting the RBD, have been shown to display neutralizing potential [1], so that this protein has been considered the most attractive target for the development of vaccines and novel specific therapies [2,3].

Although viral infections, including H1N1 influenza A and respiratory syncytial virus, have already been described as the causes of pulmonary exacerbations and the severe deterioration in lung function in patients with cystic fibrosis (CF), these patients showed no particularly severe outcomes upon SARS-CoV-2 infection [4,5,6]. A special case is that of CF patients with lung transplantation, who exhibit an increased risk of severe outcomes and intensive care following COVID-19 compared with patients in stable conditions [7,8]. An enlarged study on the general UK population, including 17 million COVID-19 patients, reported that those with immunodeficiency have an enhanced risk of severe outcomes and death [9], possibly due to weak development of the T-cell-mediated immune response upon viral infection. Hence, in March 2020, the Global Registry Harmonization Group, aimed at studying the effect of COVID-19 in patients with CF, endorsed the prioritization of SARS-CoV-2 vaccination in these patients [10]. CF lung transplant recipients (LTR) have been included within prioritization groups of the anti-SARS-CoV-2 vaccine campaign in several countries, including Italy. The Italian Ministry of Health recommended the administration of the mRNA-based vaccines BNT162b2 [11] and mRNA-1273 [12] in these patients. Both these two vaccines are able to elicit a robust humoral response, encompassing the generation of neutralizing antibodies and a remarkable cell-mediated response through CD4+ and CD8+ T-cell activation [13] and Th1 cytokine release [14]. The high efficacy of mRNA-based vaccines has been observed in different Phase 3 clinical trials, and has also been documented by real-world data, indicating that both BNT162b2 [11,15] and mRNA-1273 [12,16] have efficacy against different circulating SARS-CoV-2 variants [17,18].

Several studies on the immunological response elicited by BNT162b2 or mRNA-1273 in solid organ transplant recipients (SOTR), including heart [19], kidney [20,21] and lung [22] transplant recipients, have been carried out so far. The immune response to vaccines was found to be lower in SOTR compared with the general population, reporting reduced levels of neutralizing antibodies against the S protein and a diminished cellular response after the first, second and third doses of such vaccines [23,24]. It has been observed that the use of antimetabolite immunosuppressive therapy, high-dose corticosteroids, mammalian target of rapamycin (mTOR) inhibitors and mycophenolate mofetil (MMF) are also associated with poor responses to the BNT162b2 and mRNA-1273 vaccines, resulting in blunted humoral responses [21,23,25]. In particular, the administration of MMF has been associated with a lower antibody response to vaccination compared with other immunosuppressive therapies, including mTOR inhibitors [19]. Nonetheless, the immunogenicity of the BNT162b2 vaccine in LTR CF patients has been poorly investigated so far.

We have, hence, analyzed in this study the SARS-CoV-2 serological response to BNT162b2 vaccination in 178 patients with CF, 18 of whom were LTR. We compared the humoral response, in terms of both anti-SARS-CoV-2 S IgG and IgA neutralizing antibodies, elicited by CF patients in the presence or absence of lung transplantation. We then compared the antibody-mediated response in LTR CF patients undergoing MMF therapy with patients undergoing other immunosuppressive approaches. The safety profile of BNT162b2 was evaluated after the administration of two doses in our cohort of patients.

## 2. Results

### 2.1. Study Population

Among the 385 patients with CF aged 16 years or older constantly followed at the CF Center of Verona and Treviso, 335 (87%) accepted to receive two doses of the BNT162b2 vaccine, with an interval of 21 days. Overall, 269 of these signed the informed consent form and were included in the study. Nine patients were excluded from the analysis because they did not receive the second vaccine dose. The remaining 260 patients were therefore recruited in study.

Blood samples were drawn from all patients, but 82 patients were excluded from the antibody-response analysis because their samples failed the quality control check (i.e., presence of visible hemolysis at different time points, which is a major interferent in immunochemical assessment). Thus, a final population of 178 patients with CF (92 males, 86 females, mean age 35.1) was eligible, including 18 LTR patients (9 males, 9 females, mean age 38.8 ± 7 SD). Eight patients out of 178 reported previous infection with SARS-CoV-2, one of whom was an LTR patient, and were analyzed separately. Safety and adverse event analysis was conducted on the whole CF population that received two doses of the BNT162b2 vaccine (260 patients). Epidemiological and clinical data are summarized in Table 1. All LTR patients underwent bi-pulmonary transplant, three of whom also underwent kidney transplant, whereas one patient underwent a combined lung–liver transplant. LTR patients were a mean (SD) of 6.4 (±5.5) years from the first transplant. Immunosuppressive therapy consisted of three drugs in 17 out of the 18 LTR patients, generally including a daily mean dose of 8.2 (±6.6) mg prednisone. All LTR patients underwent calcineurin inhibitors (CNI). MMF was included in the immunosuppressive regimens in 11 out of the 18 (61%) LTR patients.

### 2.2. Serological Analysis

Immunogenicity of the BNT162b2 vaccine in patients with CF seems comparable to that observed in the general population, regardless of genotype, phenotype, therapeutic regimen or severity of the disease [26,27]. On the other hand, organ transplant recipient patients show a severely impaired antibody response to the BNT162b2 vaccine [21,23,25], in particular those undergoing MMF-dependent immunosuppression [19].

Since little is known about the humoral immune response to the BNT162b2 vaccine in LTR patients with CF, we sought to investigate this issue in our cohort of patients. Anti-SARS-CoV-2 IgA antibodies have been described as dominating the early humoral immunity response compared with IgG and IgM. Elevated levels of IgA were found in the serum, saliva and bronchoalveolar lavage fluids isolated from COVID-19 patients. Moreover, IgA antibodies were found to be more potent than IgG in neutralizing SARS-CoV-2 and higher levels of IgA have been correlated with critical illness [28,29]. Most importantly, the BNT162b2 vaccine can elicit serum levels of both anti-spike RBD IgG, and anti-spike S1 IgA [30]. Thus, we sought to investigate the ability of the BNT162b2 vaccine to elicit an IgA antibody response in CF patients. The antibody-mediated response against SARS-CoV-2 was evaluated in terms of both neutralizing IgG and IgA antibodies in serum samples collected at T0, T1 and T2 (Figure 1) from 178 patients with CF, including 18 LTR subjects.

Results indicated that the BNT162b2 vaccine already elicited a humoral immune response comparable to that observed in a general population of ostensibly healthy healthcare workers [28] after the first vaccine dose in 93.5% (IgG) and 91.5% (IgA) of non-transplanted CF patients, respectively (Table 2).

Conversely, only 0.1% and 17.6% of LTR patients showed IgG and IgA responses at T1, respectively (Table 2). In particular, at T1, non-transplanted CF patients displayed a median of 243 AU/mL of IgG, whereas LTR subjects showed a complete lack of IgG response (median 1.3 AU/mL). After 24–28 weeks from the second dose of vaccine (T2) we observed a remarkable reduction in IgG in non-transplanted CF patients, with a median of 61 AU/mL (Figure 2a). This reduction in IgG antibody response over time is in line with a recent report from another Italian cohort of CF patients [26] and has already been described in the general population as well. In fact, after three months, the IgG response elicited by two doses of the BNT162b2 vaccine decreased by 37.9% from the highest mean value in 142 seronegative healthcare professionals enrolled within the CRO-VAX HCP study [31].

Moreover, we found that anti-spike S1 IgA levels were significantly increased in non-transplanted CF patients upon vaccination both at T1 and T2, with similar median values of the Ratio of Absorbance (RoA) (4.2 and 3.5 at T1 and T2, respectively), thus well above the cut-off, which was set to 1.1 RoA. Once again, LTR subjects showed no IgA response (Figure 2b), either at T1 (median 0.5 RoA) or at T2 (0.6 RoA).

Since it has been shown that MMF therapy results in a particularly poor immune response to BNT162b2 vaccine in heart-transplanted patients, we evaluated the effect of this therapeutic regimen in CF LTR patients on the IgG and IgA antibodies response upon BNT162b2 vaccination. In the absence of MMF treatment, LTR patients showed lower levels of both IgG (3.3 and 10.9 AU/mL at T1 and T2, respectively) and IgA (0.5 and 1.2 RoA at T1 and T2, respectively) compared with non-LTR patients (Figure 2c). Nonetheless, three out of the eight LTR patients were positive for anti-spike RBD IgG in the absence of MMF treatment, whilst four patients reported positive IgA values at T2. Conversely, LTR patients undergoing MMF therapy showed a further reduction, although not statistically significant, in their humoral immune response, displaying median values of IgG of 1.5 and 1.9 AU/mL and IgA of 0.5 and 0.4 RoA at T1 and T2, respectively (Figure 2d). Moreover, none of the LTR patients under MMF treatment displayed positive values of IgG, whereas only two patients out of nine reported IgA values over the cut-off level at T2.

The antibody response to COVID-19 vaccines has been shown to vary by age and ethnicity, but not by gender, as reported in a recent large serial cross-sectional study, in which more than one million blood samples were tested [32]. Consistent with these findings, we found that median values of IgG and IgA were almost identical between males and females when tested at both T1 and T2 (Figure 2e,f), thus confirming that sex may not have a role in BNT162b2 vaccine-mediated seroprevalence, even in CF patients.

Among the 178 CF patients enrolled in our study, 9 were SARS-CoV-2 seropositive before the initiation of vaccination, including one LTR patient. However, all these patients reported mild symptoms, including a fever, sneezing, and a sore throat. None of these patients needed hospitalization or intensive care unit admission. Given the previous positivity to COVID-19 and elevated antibody titer measured already at T0, we conducted a separate analysis for these samples. As previously observed in the general population [33], we found that this subcohort of patients displayed elevated IgG levels at T0 (median 136 AU/mL) compared to those without a previous SARS-CoV-2 infection (1 AU/mL) (Figure 2g). Furthermore, patients with previous coronavirus infection displayed a significantly increased IgG antibody response to the BNT162b2 vaccine compared to those uninfected (1398 vs. 243 AU/mL and 133 vs. 61 AU/mL at T1 and T2, respectively). Similar results were found measuring IgA values in previously infected versus uninfected patients induced by the vaccine, reporting a median of 1.3 vs. 0.5 RoA at T0, 7.7 vs. 4.1 RoA at T1 and 6.3 vs. 3.4 RoA at T2 (Figure 2h).

### 2.3. Safety Report

The safety profile of the BNT162b2 vaccine was analyzed after the first and the second vaccine dose administrations in 260 pwCF. Vaccination was generally well tolerated by CF patients. Local pain at the injection site and other local reactions were mainly reported after the first administration in 177 pwCF (68%), whereas only 133 subjects (51%) experienced local pain after the second administration (Figure 3a). Pain lasted a mean of 32.5 h. Other local adverse events, including skin rashes, occurred in 8% of pwCF either after the first or second administration. One hundred and forty pwCF (55.6%) experienced systemic effects after the second vaccine dose (Figure 3a), dominated by fatigue (29%), fever (19%), headache (17%), myalgia (15%) and chills (11%) (Figure 3b). A lower percentage of pwCF (39.6%) reported the same systemic adverse events after the first vaccine dose administration (Figure 3a) but with fewer effects (Figure 3b).

## 3. Discussion

We recently described the reduced susceptibility to SARS-CoV-2 infection by CF patients, possibly attributable to decreased ACE2 expression and its mislocalization into the endoplasmic reticulum (ER) instead of the plasma membrane sustained by the loss of expression of CFTR [34]. These findings should therefore negatively affect COVID-19 progression. Notably, CF LTR patients displayed a high risk of SARS-CoV-2 infection and more severe outcomes compared to non-LTR patients [7,8]. Nevertheless, CF is still considered a comorbidity of COVID-19. In fact, CF patients with impaired lung function and CF-related diabetes have a higher risk of severe outcomes upon SARS-CoV-2 infection [7,8]. Therefore, vaccination remains an essential weapon for the primary prevention of COVID-19. In the last two years, a worldwide effort has led to the rapid development of mRNA-based anti-SARS-CoV-2 vaccines, namely BNT162b2 [11] and mRNA-1273 [12], which have demonstrated a good safety profile and immunogenicity in the general population. The BNT162b2 vaccine led to a high IgG antibody response in CF patients [26], including neutralizing antibodies [27]. Anti-SARS-CoV-2 IgA have been reported as more potent neutralizing antibodies compared with IgM and IgG, and their concentration in serum and other biological fluids correlates with the severity of COVID-19 [29,30]. Since neutralizing anti-SARS-CoV-2 IgA antibodies have not been widely investigated in CF patients yet, we tested the humoral immune response elicited by the BNT162b2 vaccine in terms of both neutralizing anti-spike (RBD) IgG and IgA antibodies.

In line with previous reports, our data confirmed that BNT162b2 stimulated a strong antibody-dependent immune response. In a study conducted on 181 seronegative healthcare workers, using the same antibody detection kits for IgG (RBD) and IgA, we observed at T1 an amount of IgG lower than that observed in the CF population (52.3 vs. 243.2 AU/mL). Analogously to what has recently been reported in the general population [31,35], we observed a decrease in IgG value after 21–24 weeks from the second vaccine dose administration (T2). On the other hand, neutralizing IgA levels at T1 were comparable to those obtained in the healthcare workers’ study (3.06 RoA, vs. 4.14). After 24–28 weeks from the first vaccine administration (T2), despite the IgG values being decreased, IgA values were instead sustained at higher levels. Since IgA antibodies against SARS-CoV-2 displayed higher neutralizing capabilities compared with IgG [29,30], our data strengthen the need for a longitudinal assessment of both IgG and IgA in order to predict the antiviral response.

Although COVID-19 does not seem to produce particularly severe outcomes in CF patients, LTR subjects have been reported to be the major morbidity and mortality risk so far. Thus, LTR patients should take the most advantage of vaccination. Nevertheless, SOTR patients generally undergo immune suppression therapies that severely impair the efficacy of vaccines [23,24]. Unfortunately, our data confirm that immune suppression can also blunt the whole humoral immune response elicited by mRNA COVID-19 vaccines in pwCF. In particular, the levels of both BNT162b2-induced anti-Spike IgG and IgA were significantly decreased in LTR patients compared with non-transplanted subjects. This is not really surprising since the type and intensity of immunosuppressive therapy are major determinants of humoral responses following SARS-CoV-2 vaccination [36]. To add to the concerns regarding the matter of the immunogenicity of COVID-19 vaccines in patients with chronic inflammatory conditions undergoing immunosuppressive treatments, a recent meta-analysis found that the rate of non-responders was around 15% in patients receiving JAK inhibitors, 6% in those receiving anti-TNF drugs, nearly 60% in those receiving anti-CD20 treatments such as rituximab, 22% in those receiving steroids, 20% in those receiving methotrexate, and around 10% in those receiving infliximab and 30% in those receiving MMF. [36]. In a cohort study published by Havlin et al. [37], for example, the total absence of an anti-SARS-CoV-2 antibodies response was observed in lung transplant recipients. Notably, even the cellular immunity appeared to be blunted, with a negative response noted in the majority of patients. Our data highlight that CF patients undergoing MMF therapy showed a lower antibody response to mRNA vaccines, especially in terms of IgG, compared with CF patients undergoing other immunosuppressive treatments. However, here we analyzed a small number of patients undergoing MMF; therefore, it would be helpful to enlarge the number of these patients in order to also confirm the effect of MMF in CF. A comprehensive analysis of COVID-19 vaccines’ immunogenic potential was conducted in a large study by Marion et al., which included nearly 900 recipients of solid organ transplants who received two doses of mRNA-based SARS-CoV-2 vaccines. Cumulatively, seropositivization occurred in less than 30% of all cases. Except for liver transplant recipients, in whom seropositivization reached 50%, the presence of anti-SARS-CoV-2 antibodies was always close to or much lower than 30% in all other solid organs recipients at 4 weeks after the second vaccine dose [38]. The potential clinical consequences of such a blunted COVID-19 vaccine immunogenicity in solid organ transplant recipients have been well described by Caillard et al., reporting the cases of 55 recipients of solid organ transplants, 52 kidney and 3 simultaneous kidney–pancreas, who developed COVID-19 despite receiving two doses of mRNA-based COVID-19 vaccines [39]. Fifteen of these 55 patients needed hospitalization for oxygen therapy, six needed intensive care and three died. Importantly, anti-SARS-CoV-2 antibodies were negative in 96% of those who underwent post-vaccination assessment [39].

Taken together, our data are, hence, consistent with the need for redefining vaccination priorities in those who are immunocompromised, thus including LTR patients, since this specific category of CF subjects may need the repeated administration of vaccine boosters, especially the new ones that have been specifically developed against the omicron variant [40]. In this perspective, we confirm the importance of a longitudinal assessment of the serological response to COVID-19 vaccination over time, encompassing the assessment of both anti-SARS-CoV-2 IgA and IgG neutralizing antibodies, since these measurements may help to the timely identification of vaccine recipients at major risk of immediate infection due to a blunted response, as well as those at enhanced likelihood of developing breakthrough infections on follow-up due to a faster decline in immunity [41].

## 4. Materials and Methods

Patients aged 16 years or older were recruited at the Cystic Fibrosis Center Regione Veneto, after informed consent was signed. During this observational study, blood tests were performed before the first dose of vaccine (T0), before administration of the second dose (T1), and 24–28 weeks after T1 (T2). A questionnaire recording local and systemic adverse events for 1 week after each administration was provided to enrolled patients at T0 and T1 and collected in the next 2–4 weeks. Serum samples were drawn after allowing blood to clot for 15 min at room temperature by centrifuging at 1000× *g* for 10 min in a refrigerated centrifuge. Semi-quantitative analysis of IgA against the S1 subunit of SARS-CoV-2 spike protein was carried out using Anti-SARS-CoV-2 QuantiVac ELISA IgA (EI 2606-9601/9620 A, Euroimmun, Lübeck, Germany), according to manufacturer’s protocol. Results were expressed as OD sample/OD standard ratio, namely RoA, with a positivity cut-off of 1.1. Neutralizing IgG against the receptor binding domain (RBD) of SARS-CoV-2 S protein were quantified using Access 2 SARS-CoV-2 IgG II kit (Beckman Coulter, Brea, CA, USA), following manufacturer’s protocol, on Access immunochemical platform (Beckman Coulter). Positivity cut-off for this assay was 30 AU/mL. Data were analyzed using SigmaPlot 14.0 (Inpixon, Palo Alto, CA, USA). Wilcoxon Signed Rank test, or the Mann–Whitney U test, were calculated in the presence of matched samples and independent samples, respectively. Categorical data were evaluated by Chi-square distribution.

## Figures and Tables

**Figure 1 ijms-24-00908-f001:**
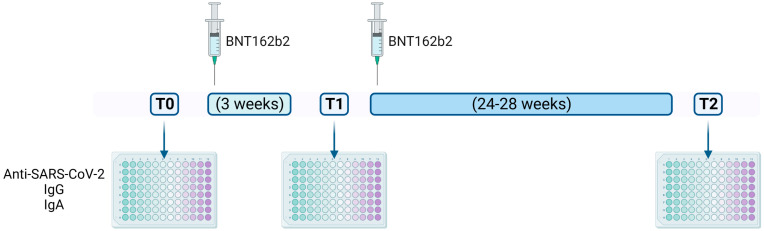
Experimental design. During this observational study, serum samples were collected (1) before the first dose of BNT162b2 vaccine (**T0**); (2) three weeks later, before the administration of the second dose (**T1**); and (3) 24–28 weeks after T1 (**T2**). Neutralizing anti-SARS-CoV-2 spike protein IgG and IgA were quantified by ELISA at T0, T1 and T2. A questionnaire recording local and systemic adverse events for 1 week after each administration was provided to enrolled patients at T0 and T1 and collected in the next 2–4 weeks.

**Figure 2 ijms-24-00908-f002:**
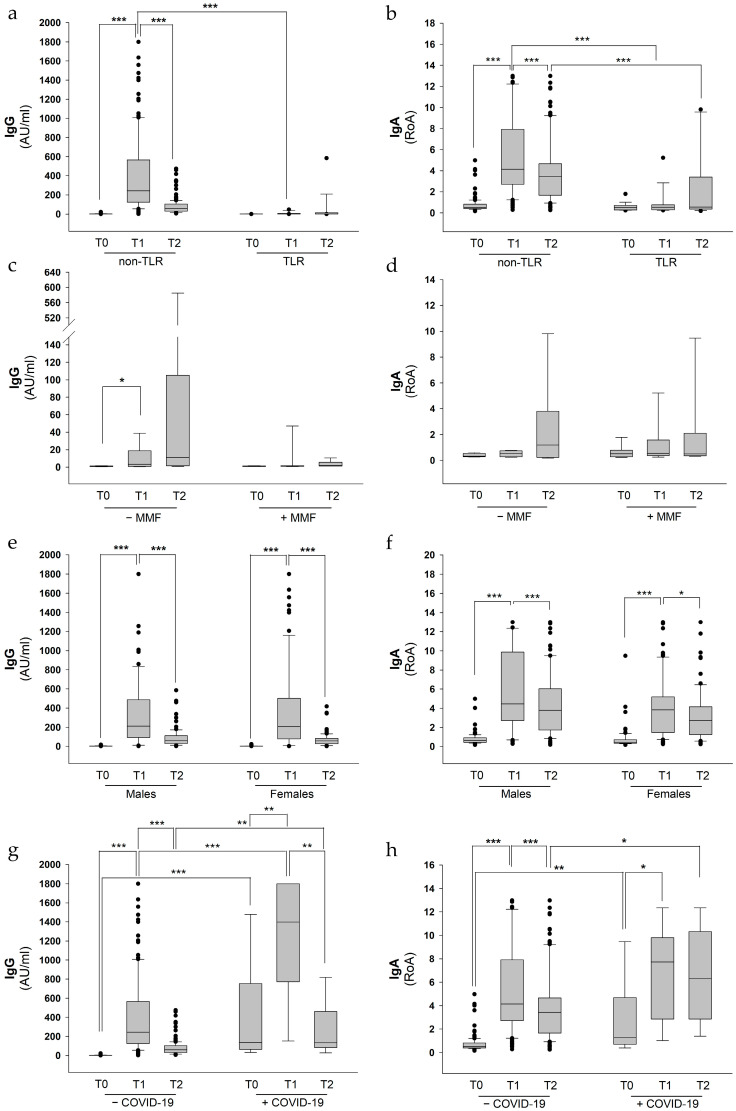
Anti-SARS-CoV-2 neutralizing IgG and IgA response in pwCF divided by categories. A total of 178 pwCF were eligible for humoral immunity assays. For data analysis, patients were then divided in different categories: non-LTR versus LTR (**a**,**b**), pwCF undergoing MMF treatment versus patients under other immunosuppressive therapies (**c**,**d**), males versus females (**e**,**f**), patients who contracted COVID-19 before vaccination versus those with no previous coronavirus infections (**g**,**h**). Wilcoxon Signed Rank test and Mann–Whitney U test were calculated for dependent and independent comparison, respectively. * *p* < 0.05; ** *p* < 0.01; *** *p* < 0.001.

**Figure 3 ijms-24-00908-f003:**
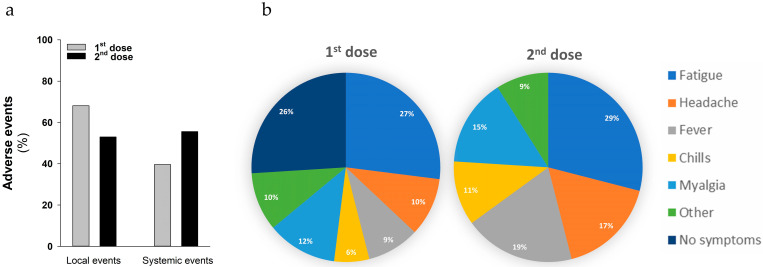
Adverse events elicited by BNT162b2 vaccine in pwCF. (**a**) Percentage of local adverse events including local pain at the injection site and rashes or systemic adverse events upon vaccination in our cohort of 260 patients after the first dose (grey histogram) and the second dose (black histogram) administration. (**b**) Main systemic adverse events reported in pwCF after first and second dose administrations of vaccine are represented as pie charts. Incidence (%) of these adverse events is indicated within the graphs.

**Table 1 ijms-24-00908-t001:** Study population: epidemiological and clinical data.

	Non-LTR	LTR	Total
N	160	18	178
M (%)	83 (51.9)	9 (50)	92 (51.7)
Previous SARS-CoV-2 infection (%)	7 (4.4)	1 (5)	8 (4.5)
Mean age (SD)	34.6 (±12.6)	38.8 (±7)	35.1 (±12.3)

**Table 2 ijms-24-00908-t002:** Percentage of pwCF reporting humoral immune response upon BNT162b2 vaccination.

	T0	T1	T2	Statistics
LTR IgG reactivity (%)	0	0.1	23.5	
Non-LTR IgG reactivity (%)	0	93.5	81.1	*p* < 0.0001 ^1^
LTR IgA reactivity (%)	0.1	17.6	52.9	
Non-LTR IgA reactivity (%)	12.4	91.5	85	*p* < 0.0001

^1.^ Chi-square test.

## Data Availability

The data presented in this study are available in the Appendix A.

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
