# Peer review of "Immunogenicity and Safety of the BNT162b2 COVID-19 Vaccine in Patients with Cystic Fibrosis with or without Lung Transplantation"

_ijms, 2023, doi:10.3390/ijms24020908_

Round 1
Reviewer 1 Report
Few minor suggestions:
1. I think the methods should be presented before the results, so that reader can follow the timeline of T0, T1, and T2 assays. I had to go back and read again as the methods section was at the end.
2. Table 2 was not aligned properly creating confusion as to which row the p-value was for. I think it's just a technical error
3. Figure 1. was hard to read and follow as the letters on the pictures were too small.
Author Response
Reviewer 1
I think the methods should be presented before the results, so that reader can follow the timeline of T0, T1, and T2 assays. I had to go back and read again as the methods section was at the end.
R1.1. We agree with this Reviewer that it should be important to present the experimental plan at the beginning of results. In the previous unedited manuscript, we followed the template of IJMS, where methods are after discussion. In order to address the request of this Reviewer we included a new Figure 1, in which we clearly explained the experimental plan and the meaning of T0, T1 and T2.
2. Table 2 was not aligned properly creating confusion as to which row the p-value was for. I think it's just a technical error.
R1.2. We modified Table 2 according to the Reviewer suggestion.
3. Figure 1. was hard to read and follow as the letters on the pictures were too small.
R1.3. We agree with this Reviewer’s comment. We have now enlarged Figure 1 and we modified the text dimension of legends in order to make it more readable.
Reviewer 2 Report
In this manuscript, authors investigate the immunogenicity and safety of SARS-CoV-2 mRNA vaccines in patients with cystic fibrosis (CF) with or without lung transplants (LTR). This study adds on to various other studies which investigate the immunogenicity of SARS-CoV-2 vaccines in patients with comorbidities, organ transplants and immunosuppression. Here they observed that vaccine induced IgG and IgA antibody response in CF patients was similar to those observed in general population but significantly impaired in LTR patients.
Study is well designed, and endpoints are measured and interpreted accurately. I have following comments
· One of the observations that stands out from this manuscript is that there is remarkable reduction in IgG response after second dose of vaccine which is contrary to response in general population. Can authors discuss what could be the reasons behind this.
· Add description of T0, T1 and T2 and the timings of serum collection in the beginning of manuscript so it’s easy to follow and reader don’t have to go to methods for these details.
· Line 153- Add full form of RoA at least once.
· Table 1- write SARS in capital letters.
Author Response
Reviewer 2
In this manuscript, authors investigate the immunogenicity and safety of SARS-CoV-2 mRNA vaccines in patients with cystic fibrosis (CF) with or without lung transplants (LTR). This study adds on to various other studies which investigate the immunogenicity of SARS-CoV-2 vaccines in patients with comorbidities, organ transplants and immunosuppression. Here they observed that vaccine induced IgG and IgA antibody response in CF patients was similar to those observed in general population but significantly impaired in LTR patients.
Study is well designed, and endpoints are measured and interpreted accurately.
We thank the Reviewer for her/his kind words of appreciation.
I have following comments
- One of the observations that stands out from this manuscript is that there is remarkable reduction in IgG response after second dose of vaccine which is contrary to response in general population. Can authors discuss what could be the reasons behind this.
R2.1. As observed by the Reviewer, after second dose of vaccine we observed a reduction of IgG response at T2. Since T2 was planned after 24-28 weeks (almost 6 months), our data are in line with already previously reported observations on time-dependent antibody titre decline upon BNT162b2 administration. For instance, Favresse et al. described that the maximal antibody response upon BNT162b2 vaccine in the general population was reached between days 28 and 42. Afterward, a continuous decrease was observed until day 90. The estimated half-life of antibodies observed from data collected until 90 days post-vaccination for seronegative participants was 55 days indeed (Favresse et al., Emerg Microbes Infect. 2021 doi: 10.1080/22221751.2021.1953403.). Similar results were reported by Mileto et al. in another study (Mileto et al. Emerg Microbes Infect. 2021doi: 10.1080/22221751.2021.2004866). Nevertheless, according to the Reviewer suggestion, we further highlighted this point, citing these two papers as supporting reports (please see lines 174-181 and 333-335 in the revised manuscript).
Add description of T0, T1 and T2 and the timings of serum collection in the beginning of manuscript so it’s easy to follow and reader don’t have to go to methods for these details.
R2.2. We totally agree with the Reviewer’s comment. We have added new Figure 1 explaining the experimental design, clarifying the meaning of T0, T1 and T2.
- Line 153- Add full form of RoA at least once.
R2.3. We add full form of RoA in line 183 (former line 153) in the revised manuscript, accordingly.
- Table 1- write SARS in capital letters
R2.4. We have modified the text of Table 1 accordingly.
Reviewer 3 Report
1. The article is well written and well constructed, which is impressive selection criteria and an important topic to be discussed. , anyhow, Consent form and ethical approvals should be obtained before publishing this data. 2. Table 2 should be restructured and better represented. I should be easy and self explanatory. 3. If the error bars in the figures are excessively large, I would have a professional statistician review the results. , 4. Similarly, in figures 1c and 1d, the results for +MMF and -MMF, IgG and IgA are not significant and should be avoided. 5. in the last figure, what there are no "No symptoms" count in the figure? the comparison is 134 vs 100 I believe? overall I would accept the article if statistician scores it high.
Author Response
Reviewer 3
1. The article is well written and well constructed, which is impressive selection riteria and an important topic to be discussed.
We thank the Reviewer for appreciating our work.
anyhow, Consent form and ethical approvals should be obtained before publishing this data.
R3.1. We totally agree with the Reviewer. As indicated in the journal template, we described ethical approvals below the Material and Methods section. We have already stated that “The study was conducted in accordance with the Declaration of Helsinki, and approved Ethics Committee of the Azienda Ospedaliera Universitaria Integrata di Verona (protocol code CRCFC-VACMRNA051, approval number CE3249)” and Informed Consent Statement: “Informed consent was obtained from all subjects involved in the study” (please see lines 415-419).
2. Table 2 should be restructured and better represented. I should be easy and self explanatory.
R3.2. We modified the Table 2 according to the suggestions from Reviewer 1 and Reviewer 3.
3. If the error bars in the figures are excessively large, I would have a professional statistician review the results.
R3.3. All data were analyzed by a professional statistician (Dr. Gloria Tridello).
4. Similarly, in figures 1c and 1d, the results for +MMF and -MMF, IgG and IgA are not significant and should be avoided.
R3.4. We agree with this Reviewer that there is not an evident difference between patients undergoing MMF and patients undergoing other immunosuppressors. However, we believe that it should be important to show and comment data from CF patients under immunosuppressive therapy. In any case, our data suggest that there is a trend of better response to BNT162b vaccine in CF patients without MMF. Since data from LTR subjects were obtained from a small number of cases, it should be necessary to follow a larger cohort of patients with similar condition in further studies. We added this point in the discussion (please see lines 362-364).
5. in the last figure, what there are no "No symptoms" count in the figure? the comparison is 134 vs 100 I believe?
R3.5. The new Figure 3a describes percentage of adverse events, it does not show raw data. The same is for panel b, where the numbers inside the pie charts describe the percentage of patients reporting each adverse event. In the case of “no symptom”, we reported that after the second dose of vaccine all patients reported at least one symptom, described in the pie chart.